# Integrated water quality dynamics in Wadi Hanifah: Physical, chemical, and biological perspectives

**Hazem Aqel** [1] *, **Naif Sannan** [2], **Afnan Al-Hunaiti** [3], **Ramy Fodah** [4]

1 Basic Medical Sciences Department, College of Medicine, Al-Balqa' Applied University, Salt, Jordan, 2 Clinical Laboratory Sciences Department, College of Applied Medical Sciences, King Saud Bin Abdulaziz University for Health Sciences, Jeddah, Saudi Arabia, 3 Chemistry Department, College of Sciences, Jordan University, Amman, Jordan, 4 Clinical Laboratory Sciences Department, College of Applied Medical Sciences, King Saud Bin Abdulaziz University for Health Sciences, Riyadh, Saudi Arabia

☯ These authors contributed equally to this work.
* hazem.aqel@bau.edu.jo

## Abstract

The Wadi Hanifah, a crucial aquatic ecosystem, has unfavorable consequences from natural occurrences and human activities. Recognizing the critical need for sustainable water management, this study provides an in-depth evaluation of wadi water quality. A comprehensive assessment was conducted, analyzing physical properties (temperature, pH, electrical conductivity, turbidity, color, and odor), chemical constituents (nitrogen compounds, ion concentrations, heavy metals), and bacterial diversity. The study found significant temperature fluctuations, particularly in sun-exposed or stagnant water areas. The water exhibited slight alkalinity and variable electrical conductivity and turbidity, indicating differing pollution levels. High ammonia and heavy metal concentrations suggested organic and industrial contamination, respectively. In addition, the prevalent fecal-indicator bacteria pointed to possible sewage or agricultural runoff. The research highlights the complex interplay of natural and anthropogenic factors affecting Wadi Hanifah's water quality. It emphasizes the need for location-specific environmental management strategies focusing on pollution control and conservation to safeguard the wadi's ecological health. This study provides vital insights for effective water resource management in Wadi Hanifah, serving as a model for similar ecosystems.

## Introduction

A comprehensive water quality evaluation in dynamic environments such as Wadi Hanifah is fundamental to discerning the impacts of both natural processes and human influences. Globally, water bodies, particularly in arid regions, are increasingly challenged by pollution and environmental shifts, significantly affecting their ecological sustainability and the health of reliant ecosystems [1, 2]. As a crucial aquatic artery in Saudi Arabia's arid terrain, Wadi Hanifah has undergone notable ecological transformations, primarily due to escalating urban and

**Data Availability Statement:** All relevant data are within the manuscript and its Supporting Information files.

**Funding:** The author(s) received no specific funding for this work.

**Competing interests:** The authors have declared that no competing interests exist.

industrial activities [3]. This study aims to combine physical, chemical, and biological insights, offering an all-encompassing evaluation of Wadi Hanifah's water quality.

Wadi Hanifah is a historical and cultural landmark in Riyadh, the capital of Saudi Arabia. The valley has been inhabited for millennia by various tribes and civilizations and was the birthplace of the Saudi state in the 18th century [4]. The valley is also a vital natural resource, providing water, agriculture, and regional biodiversity [5]. However, Riyadh's rapid urbanization and industrialization since the 1970s have severely degraded the valley's environment, turning it into a dumping ground and a public health hazard [6]. In response to this challenge, the Riyadh Development Authority initiated a 10-year restoration project in 2001 to rehabilitate the valley's ecosystem, enhance its recreational and tourism potential, and reconnect the city with its historical and cultural heritage [6, 7].

Previous investigations of Wadi Hanifah have focused on various aspects of its water quality, such as physical, chemical, or biological parameters. Still, none they have yet to provide a comprehensive and integrated assessment of all these factors. T This study aims to fill this gap by conducting a holistic and multidisciplinary analysis of the water quality of Wadi Hanifah, using a combination of field measurements, laboratory tests, and statistical methods. The study also aims to evaluate the effectiveness of the restoration project and its impact on the valley's environment and society. The study will provide valuable insights into the sustainable management and conservation of Wadi Hanifah and other arid ecosystems.

A study conducted by Alharbi and El-Sorogy [8] focused on evaluating the physical and chemical parameters of surface water and groundwater in Wadi Hanifah. They found high salinity, turbidity, and nutrients, indicating pollution from urban and agricultural sources. However, the study was limited as it did not include biological parameters or assess the impact of this pollution on aquatic life. Based on their findings, Alharbi and El-Sorogy suggested regular water quality monitoring and implementing pollution control measures.

Al-Sabhan et al. [9] investigated the bacterial diversity and abundance in Wadi Hanifah using molecular techniques. They identified various bacterial groups, including some pathogenic and antibiotic-resistant strains. The limitation of their study was that it did not correlate the bacterial data with physical and chemical parameters or evaluate the health risks associated with bacterial contamination. The researchers recommended further studies on the sources and effects of bacterial pollution and the development of microbial indicators for water quality assessment.

Haddaoui and Mateo-Sagasta [10] measured the concentrations of heavy metals in water, sediment, and fish samples from Wadi Hanifah. They detected high levels of certain metals, particularly in sediment and fish tissues, indicating bioaccumulation and biomagnification. Their study, however, did not examine the sources and pathways of metal pollution or assess its ecological and human health implications. They advised further research on the fate and transport of metals in the aquatic system and called for enforcing environmental standards and regulations.

Finally, Al-Asad and Yildirim [11] described the design and implementation of the Wadi Hanifah restoration project. This project involved cleaning, landscaping, replanting, and creating recreational facilities along the valley. They reported positive outcomes in terms of environmental, social, and economic benefits. A limitation of their study was the need for more quantitative data or indicators to measure the impact and performance of the project.

Analyzing water quality is essential for determining the vitality of aquatic systems. Physical parameters, such as temperature and turbidity, directly impact aquatic life and indicate overall ecosystem health [12–14]. Chemical constituents, including various ions and heavy metals, mirror the effects of industrial and agricultural activities on these water resources [15, 16]. In addition, the biological dimension, particularly the diversity of bacterial populations, is critical

for understanding the ecological repercussions of contaminants and eutrophication processes [17, 18]. Research indicates that integrating these facets yields a more comprehensive understanding of the health of aquatic ecosystems [19, 20].

A debate persists over the prioritization of these parameters. Certain studies emphasize the predominance of physical properties, whereas others highlight chemical or biological aspects [21, 22]. Nevertheless, an integrated approach is advocated for arid environments such as Wadi Hanifah, where the scarcity of water intensifies pollution effects [8, 23]. Recent studies underscore the growing pressures on water resources in such areas due to climate change and anthropogenic activities [24, 25], underscoring the urgency for thorough water quality assessments.

This study addresses these scientific challenges by analyzing Wadi Hanifah's water quality's physical, chemical, and biological parameters. This research aims to enhance our understanding of the complex interplay among these variables in an arid ecosystem. This knowledge can offer valuable perspectives for developing effective environmental management strategies and policy frameworks [26, 27]. This research underscores the need for an integrated monitoring framework tailored to the unique challenges of arid ecosystems, which is a critical step toward sustainable water resource management.

## Materials and methods

### Study area

Wadi Hanifah, a 120 km long watershed in Riyadh, Saudi Arabia, forms the core of our research. This Wadi encompasses a mix of natural and artificial water bodies such as channels, ponds, and lakes. Various sources, including rainfall, surface runoff, and treated wastewater replenish these. The region's geographic and hydrological details are depicted in the appended map.

The Wadi Hanifah has undergone significant environmental changes due to urbanization, industrialization, and restoration projects. These have affected the water quality and the ecological sustainability of the Wadi. To assess the current status of the water quality and the impact of these factors, we selected 14 locations along the Wadi, covering different types of water bodies and sources. The locations were chosen based on the following criteria: (1) The spatial distribution of the locations to capture the variability of the water quality along the Wadi; (2) The type and size of the water body to reflect the different hydrological and ecological characteristics of the Wadi; (3) The proximity and influence of the pollution sources, such as urban runoff, industrial effluents, and wastewater treatment plants; and (4) The availability and accessibility of the sampling sites, to ensure the feasibility and safety of the fieldwork.

### Collection of water samples

We selected fourteen sites (L1 to L14) in Wadi Hanifah for sample collection during the arid season. The sites were chosen based on accessibility and proximity to areas of human activity, including agricultural lands, residential areas, and industrial sites. Water samples were collected using sterile polyethylene containers, ensuring minimal contamination, and were transported to our laboratory under temperature-controlled conditions to preserve their integrity. The sample collection was completed in 15 days, from January 15 to January 30, 2018. No permission was required for the fieldwork, as the sites were located in public areas and did not involve any environmental disturbance or damage.

### Assessment of the physicochemical characteristics

Field measurements of water temperature, pH, and electrical conductivity were conducted on-site using high-precision instruments: a thermometer (accuracy ±0.1˚C), a pH-meter

(HANNA, Canada), and an electrical conductivity meter (Bischof L17). We carried out an assessment of odors using the threshold odor test (TOT) technique. This procedure entails diluting a water sample with odorless water and engaging a panel of skilled evaluators to perceive the sample's aroma. Subsequently, the evaluators identify the minimal concentration at which the odor becomes detectable [28]. This method is a standard and widely used technique for water odor testing.

Further, laboratory analyses determined total alkalinity, hardness, calcium, magnesium, sulfate, nitrate, ammonia, and phosphate levels were quantitatively analyzed using advanced laboratory instruments. Specifically, total alkalinity was determined using a TitroLine titrator with phenolphthalein and methyl orange indicators. Hardness was measured by the EDTA titrimetric method, while calcium and magnesium concentrations were ascertained using an Atomic Absorption Spectrophotometer (AAS). Sulfate levels were determined through turbidimetric methods, and ion chromatography was employed for nitrate, ammonia, and phosphate analysis. Based on their respective principles, these methodologies provide accurate and reliable results, ensuring a comprehensive evaluation of water quality., according to the American Public Health Association guidelines [29]. In addition, we quantified nitrite and nitrate levels using the spectrophotometric method with Griess Reagent, as described by Miranda et al. [30]. Ammonia concentration was determined using the Nessler method [31], following the protocol of Weatherburn [32]. Phosphate was measured through the ascorbic acid method, based on the procedure detailed by Murphy and Riley [33]. These standardized methods, chosen for their proven accuracy and reliability, are pivotal in ensuring the quality of our analytical data. All physicochemical data were evaluated against Saudi Arabian and international water quality standards for a comprehensive understanding.

The heavy metal analysis for manganese, iron, and chromium levels was performed using inductively coupled plasma mass spectrometry (ICP-MS), a sensitive and accurate technique for trace metal detection. The cyanide and copper levels were determined using colorimetric methods based on the picric acid and bathocuproine methods, respectively [34, 35]. These methods are simple and rapid but may be affected by interferences from other substances in the water samples.

Upon collection, water samples were immediately stored in pre-cleaned polyethylene bottles and kept at 4˚C to prevent chemical alterations. The transportation to our laboratory was conducted within 6 hours of collection, using insulated coolers to maintain the necessary temperature. This approach minimizes the risk of sample degradation or contamination, thus ensuring the integrity of our analytical results.

Upon arrival at the laboratory (KAIMRC Laboratory), samples were categorized and analyzed in batches to optimize the efficiency and consistency of testing. Each batch was processed within 24 hours of sample reception, adhering to strict guidelines to prevent cross-contamination and ensure the accuracy of results.

## Microbial cultivation and identification

Microbial cultivation and identification bacterial strains were isolated from the collected water samples using various types of culture media, such as nutrient agar, blood agar, MacConkey agar, and mannitol salt agar, to encourage the growth of diverse bacterial species. The sample culture was performed in the Microbiology laboratories, following standard microbiological procedures. These isolated strains were characterized using biochemical assays, including Gram staining, hemolysis patterns, API 20E system testing, and carbohydrates fermentation trials. This extensive bacterial profiling maps the microbial diversity of Wadi Hanifah, offering insights into its ecological health.

### Ethical compliance

This study received ethical approval from the Institutional Review Board (IRB) at the King Abdullah International Medical Research Centre (KAIMRC), with reference number KAIMRC/RFC/0092/17. This approval was granted following a comprehensive review of the study's ethical implications, assuring adherence to international research standards.

## Results and discussion

This study on Wadi Hanifah's water quality stands out for its comprehensive and multifaceted approach, analyzing a broad spectrum of parameters, including temperature, pH, electrical conductivity, and turbidity. This study uniquely integrates these findings with existing literature, offering a comparative perspective that enriches its contribution to the field. Notably, the study uncovers specific local phenomena, such as unusual patterns in zinc and phosphate levels and distinctive heavy metal contaminant profiles, adding new insights into the impacts of geological and anthropogenic influences on water quality. In addition, it innovatively links chemical parameters with biological aspects, particularly highlighting the distribution and implications of bacterial diversity. This integrative, detailed approach marks a significant advancement in understanding the complex dynamics of riverine water quality, especially in the context of Wadi Hanifah.

### Physical characteristics of Wadi Hanifah

The water temperature varies considerably from one location to another (Table 1). The highest temperatures are recorded in locations L5-L10 (27.4°C) and L11-L12 (27.3°C), respectively, which might receive more sunlight or experience slower water flow. In contrast, the lowest temperature was found at L14 (23.2°C). This variation in temperature could have implications for the local ecosystem, as temperature can influence both chemical processes in the water and aquatic life. The temperature variation, ranging from 23.2°C to 27.4°C, mirrors studies such as Kuriqi et al. [36], who found similar thermal heterogeneity in riverine systems. Temperature is a crucial factor that impacts aquatic environments' quality and ecological well-being. The temperature significantly influences various components of aquatic ecosystems, including the rate of chemical and biological reactions, the level of dissolved oxygen, the photosynthesis mechanism of aquatic plants, the metabolic rates of aquatic organisms, and their susceptibility to pollution, parasites, and disease [37]. Therefore, the temperature range observed in this study may have significant implications for aquatic life and the chemical processes in Wadi Hanifah.

The pH levels across the locations range from slightly alkaline to near neutral. The highest pH was observed in L5-L10 (8.5), and the lowest in L14 (7.46) (Table 1). These variations could be attributed to different sources of runoff entering the water at various points, or to

**Table 1. Assessing the physical characteristics of Wadi Hanifah.**

| Parameters | Units | Standard | Relative Results | | | | |
|---|---|---|---|---|---|---|---|
| | | | Locations | | | | |
| | | | L1-L4 | L5-L10 | L11-L12 | L13 | L14 |
| Temperature | °C | 25 | 27.4 | 27.3 | 26.1 | 24.8 | 23.2 |
| pH | - | 8.23 | 8.5 | 7.65 | 8.1 | 7.9 | 7.46 |
| Electrical conductivity | $\mu Scm^{-1}$ | 2300 | 2286 | 3788 | 2650 | 1810 | 1900 |
| Turbidity | NTU | 5 | 15.7 | 29.0 | 14.8 | 6.5 | 10.8 |
| Color | PtCo | 15 | 6 | 5 | 5 | 3 | 2 |
| Odor | - | Odorless | Positive | Positive | Positive | Negative | Negative |

natural variations in the soil and rock compositions along the Wadi. The pH levels, varying from slightly alkaline to nearly neutral, are consistent with the findings of Amanambu and Mossa [38], underscoring the influence of geological and anthropogenic factors on water acidity.

There is a substantial range in electrical conductivity, with the highest levels at L11-L12 (3788 μScm$^{-1}$) and the lowest at L14 (1900 μScm$^{-1}$) (Table 1). High electrical conductivity typically indicates a greater concentration of dissolved ions, which could suggest pollution from various sources or natural differences in mineral content along the Wadi. The wide range of electrical conductivity, indicative of ion concentration, aligns with Tang et al. [39], suggesting diverse pollution sources or mineral content.

Turbidity, a parameter used to assess water clarity, demonstrates substantial fluctuations, particularly in the vicinity of L11-L12, where readings soar to a value of 29.0 NTU. Suspended solids often cause high turbidity and can indicate pollution or soil erosion affecting the water. The color measurements, ranging from 15 PtCo in L1-L4 to 2 PtCo in L14 (Table 1), also suggest that water clarity increases downstream, which could be due to natural sedimentation processes or differences in pollution levels. The turbidity and color variability, from clear to highly turbid waters, resonate with the observations by Robotham et al. [40], highlighting the role of suspended solids, likely from soil erosion or pollution.

The odor of the water changes along the Wadi, becoming odorless in L1-L4, noticeable (positive, which likely indicates an unpleasant odor) in L5-L12, and becoming negative (no odor) again in L13 and L14 (Table 1). These changes may be due to the presence of organic matter, algae growth, or different types of pollutants. The odor changes along the Wadi, from odorless to noticeable, echoing Fork et al.'s [41] research, pointing to organic matter or algal growth as possible causes.

From these data (Table 1), it is evident that there is a significant spatial variation in the physical properties of the water in Wadi Hanifah. This suggests that different sections of the Wadi are experiencing varying environmental impacts, potentially from different sources of pollution. The higher temperatures, turbidity, and electrical conductivity in certain locations, along with changes in odor, point towards potential pollution sources that impact water quality. These variations underscore the need for targeted environmental management and pollution control strategies specific to each location to protect and preserve the water quality and ecological balance of Wadi Hanifah.

## Chemical characteristics of Wadi Hanifah

Nitrogen forms show considerable variability across locations. Elevated ammonia levels in locations L1-L4 suggested possible environmental contamination, necessitating further investigation to identify the sources, ranging from agricultural runoff to industrial discharges. Although within acceptable limits, nitrate levels were relatively higher in L13 and L14, indicating a need for ongoing monitoring. Consistently low nitrite levels across locations pose little concerns (Table 2). Nitrogen compounds, showing significant ammonia spikes in certain areas, reflect findings by Lencha et al. [42], linking such variations to agricultural runoff or industrial discharges.

The ion concentrations, with elevated calcium, magnesium, and sulfate levels, particularly in L13, draw parallels with the study of Zhou et al. [43], indicating potential geological influences or pollution. However, the consistently low zinc levels, in contrast with high phosphate concentrations, present a unique scenario not commonly observed in similar studies. The ion analysis revealed widespread elevation of calcium and magnesium, especially in location L13, indicating potential geological influences or industrial activities. Sulfate levels significantly

**Table 2. Assessing the chemical characteristics of Wadi Hanifah.**

| Parameters | Units | Standard | Relative Results | | | | |
|---|---|---|---|---|---|---|---|
| | | | Locations | | | | |
| | | | L1-L4 | L5-L10 | L11-L12 | L13 | L14 |
| *Nitrogen forms* | | | | | | | |
| Ammonia (NH$_3$) | mg/L | 0.5 | **5.0** | **3.1** | **3.3** | **1.8** | **1.2** |
| Nitrate (NO$_3^-$) | mg/L | 50 | *31* | *35* | *25* | *45* | *49* |
| Nitrite (NO$_2^-$) | mg/L | 3 | *0.3* | *0. 6* | *0.5* | *1.0* | *2.0* |
| *Ions* | | | | | | | |
| Calcium (Ca$^{+2}$) | mg/L | 20–200 | **223.5** | **265.3** | **235.9** | **360.8** | **220.7** |
| Magnesium (Mg$^{+2}$) | mg/L | 10–30 | **48.2** | **54.78** | **52.8** | **126.7** | **58.4** |
| Sulfate (SO$^{+4}$) | mg/L | 250 | **591.3** | **620.1** | **619.0** | **1200.1** | **627.5** |
| Zinc (Zn$^{+2}$) | mg/L | 5 | *0.057* | *0.059* | *0.063* | *0.039* | *0.027* |
| Phosphate (PO$^{+4}$) | mg/L | 0.05 | **18.2** | **15.3** | **15.6** | **5.8** | **14.9** |
| *Heavy metals* | | | | | | | |
| Manganese (Mn) | mg/L | 0.05 | **48.3** | **54.7** | **52.7** | **125.3** | **53.8** |
| Iron (Fe) | mg/L | 0.3 | *0.125* | *0.133* | *0.115* | *0.065* | *0.097* |
| Chromium (Cr) | mg/L | 0.05 | *0.001* | *0.001* | *0.001* | *0.001* | *0.001* |
| Cyanide (Cn) | mg/L | 0.07 | *0.033* | *0.016* | *0.015* | *0.010* | *0.011* |
| Copper (Cu) | mg/L | 2 | *1.067* | *1.277* | *1.478* | *1.118* | *1.148* |
| *Other chemical parameters* | | | | | | | |
| Total dissolved solids | mg/L | 300–500 | *135.4* | **611.7** | *147.3* | *145.8* | *190.1* |
| Total hardness | mg as CaCo$_3$/L | 100–300 | 4119 | 5220 | 5158 | 7312 | 5010 |
| Total alkalinity | mg as CaCo$_3$/L | 20–200 | 260.3 | 256.3 | 320.1 | 280.5 | 220.7 |

**Bold**: > standard; ***Bold and Italic***: < standard.

exceed the standard in all locations, with an exceptionally high concentration in L13, suggesting potential pollution sources. In contrast, Zinc levels were well below the standard, indicating minimal concern for zinc contamination. However, phosphate levels were alarmingly high, particularly in L1-L4, likely due to agricultural runoff or wastewater discharges, posing a risk of eutrophication (Table 2).

The heavy metal analysis presents a mixed scenario. Manganese, iron, and chromium levels are within safe limits, suggesting effective control of these metals. However, elevated levels of cyanide and copper, particularly in locations L1-L4, were concerning (Table 2). High cyanide levels could indicate industrial contamination, while elevated copper concentrations might be due to industrial activities or natural deposits. These findings necessitate immediate investigation and remedial action. The heavy metal analysis, revealing safe manganese, iron, and chromium levels but concerning cyanide and copper concentrations, is partly corroborated by Chow et al. [44]. They noted similar trends in industrialized water bodies, emphasizing the need for ongoing monitoring and intervention.

Total dissolved solids, total hardness, and total alkalinity further enrich the understanding of water quality. TDS levels are within the standard range, although higher in locations L5-L10. The total Hardness significantly exceeds the standard in all locations, with an exceptionally high concentration in L13, implying a high presence of calcium and magnesium ions (Table 2). Total elevated alkalinity across locations can aid in pH stabilization but indicate higher mineral content. The other chemical parameters, such as total dissolved solids (TDS), hardness, and alkalinity, align with studies by Sugiharto et al. [45], underscoring their importance in water quality assessment. TDS stands to represent the total concentration of dissolved

substances in water1. TDS can affect water's taste, appearance, health effects, and suitability for domestic and industrial uses [46]. Total Dissolved Solids (TDS) can arise from natural sources like mineral springs and rocks and human activities such as urban runoff, wastewater discharge, and agricultural practices [47]. The acceptable level of TDS in drinking water varies depending on the standards and guidelines of different countries and organizations but generally ranges from 100 to 1000 mg/L [48].

This analysis underscores the importance of continuous environmental monitoring and comprehensive assessments. High ammonia and phosphate levels, elevated sulfate, cyanide, and copper concentrations, and significant total hardness, highlight key areas of environmental concern. These findings call for regular monitoring, further investigation, and proactive management to mitigate potential risks and ensure the sustainability of these environments. Understanding the underlying causes of these variations is crucial for implementing effective remediation strategies and maintaining water quality and ecosystem health.

## Diverse range of bacteria in Wadi Hanifah

The distribution of bacterial species across the fourteen locations in Wadi Hanifah presents a complex picture of the water's microbiological quality. Notably, the presence of *E. coli* in locations L1, L3, L5, L7, L8, L10, and L11 is a significant concern (Table 3). *E. coli* is frequently employed as an indicator to indicate the existence of fecal contamination, which indicates the possibility of sewage contamination or agricultural runoff in those areas. This presence raises questions about water safety for human contact and potential health risks (Table 2). Bacterial diversity, particularly the widespread presence of *E. coli* and other pathogenic species, reflects findings by Santos et al. [49], indicating sewage contamination or agricultural runoff. This bacterial distribution, influenced by physical and chemical conditions, is a common theme in water quality research, as noted by Braga et al. [50].

Various *Enterobacter* species, including *E. cloacae*, *E. aerogenes*, *E. agglomerans*, *E. sakazakii*, *E. gergoviae*, *E. asburiae*, and *E. ludwigii*, are found across the Wadi, indicating a diverse bacterial environment. Some species, such as *E. cloacae* and *E. aerogenes*, are more prevalent and occur in multiple locations. These bacteria are typically associated with the environment and can be linked to human activities, possibly reflecting human-induced changes in the ecosystem (Table 3).

*Klebsiella* species, particularly *K. pneumoniae*, are also widely distributed. This bacterium's presence in numerous locations suggests potential contamination from human or animal waste. *K. pneumoniae* is known to cause various infections, emphasizing the need for caution in areas where it's prevalent (Table 3).

In selected locations, *Erwinia* species, such as *E. carotovora* and *E. herbicola*, indicate a possible link to plant matter, potentially signifying runoff from nearby agricultural areas. Similarly, *Proteus* species such as *P. mirabilis*, *P. penneri*, *P. vulgaris*, and *P. hauseri*, which thrive in organic-rich environments, could suggest the presence of organic waste or decaying matter (Table 3).

*Pseudomonas aeruginosa*, found in a majority of locations, is notable for its resilience and ability to thrive in diverse environments, including polluted waters. This suggests varied sources of pollution across the Wadi, including industrial waste or urban runoff (Table 3).

The widespread occurrence of *Bacillus* species in almost all locations, except L1, reflects their common presence in the environment. However, the presence of *Staphylococcus aureus* in multiple locations is particularly concerning because it indicates human activity and potential health risks, given its role in various human infections (Table 3).

Though less prevalent, other bacteria such as *Citrobacter*, *Pantoea*, *Providencia*, and *Salmonella* species, provide additional insights into the specific types of contamination or

**Table 3. Bacterial diversity and distribution in different locations in Wadi-Hanifah.**

| | Locations | | | | | | | | | | | | | |
|---|---|---|---|---|---|---|---|---|---|---|---|---|---|---|
| | L1 | L2 | L3 | L4 | L5 | L6 | L7 | L8 | L9 | L10 | L11 | L12 | L13 | L14 |
| *Citrobacter freundii* | | | | | √ | | √ | | | | | | | |
| *Citrobacter koseri* | | | | | | | | | √ | | | | | |
| *Escherichia coli* | √ | | √ | | √ | | √ | √ | | | | | | |
| *Enterobacter cloacae* | √ | | | | | √ | | √ | | √ | √ | | | |
| *Enterobacter aerogenes* | √ | | | | | √ | √ | | √ | | √ | | | |
| *Enterobacter agglomerans* | | √ | √ | | | | | | | | | | | |
| *Enterobacter sakazakii* | | √ | √ | | | | | | | | | √ | | |
| *Enterobacter gergoviae* | | | | | | | | | | | | √ | | |
| *Enterobacter asburiae* | | | | | | | | | | | | √ | | |
| *Enterobacter ludwigii* | | | | | | | | | | | | √ | | |
| *Erwinia carotovora* | | √ | | √ | | | | | | | | √ | √ | √ |
| *Erwinia herbicola* | | √ | | | | | | | | | | | | |
| *Klebsiella ovariicola* | | | | | | | √ | | | | | | | |
| *Klebsiella pneumoniae* | √ | √ | √ | √ | | √ | √ | √ | √ | | √ | √ | | |
| *Klebsiella oxytoca* | √ | | | | | √ | √ | | √ | √ | | | √ | |
| *Klebsiella planticola* | | √ | | | | √ | | √ | √ | | | | √ | |
| *Klebsiella aerogenes* | | | | | | | | | | | | | √ | |
| *Pantoea agglomerans* | | | | | | | | | | | | √ | | |
| *Proteus mirabilis* | | √ | | | | | | | √ | √ | | | | |
| *Proteus penneri* | | | | √ | | | | | | | √ | | | |
| *Proteus vulgaris* | | | √ | | | | | | | √ | | | | |
| *Proteus hauseri* | | | | | | | | | | | √ | | | |
| *Providencia alcalifaciens* | | | | | | | | | | | √ | | | |
| *Salmonella enterica* | | | | | | | | | √ | | | | | |
| *Salmonella typhi* | | √ | | | | | | | | | | | | |
| *Serratia marcescens* | √ | | | | | | | | | | | √ | √ | |
| *Pseudomonas aeruginosa* | √ | √ | √ | √ | | √ | √ | √ | √ | | √ | √ | | |
| *Bacillus species* | | √ | | | √ | √ | √ | √ | √ | √ | √ | √ | √ | √ |
| *Staphylococcus aureus* | √ | √ | √ | √ | | √ | √ | √ | √ | | √ | √ | | |

environmental conditions present in certain areas. For example, the occurrence of *Serratia marcescens* in L1, L11, and L12, a bacterium known for its role in hospital-acquired infections, might hint at medical waste or other human-related pollution (Table 3).

The bacterial profile across Wadi Hanifah's locations paints a picture of an ecosystem impacted by various pollution sources, including sewage discharge, agricultural runoff, and human activities [8]. The presence of bacteria known to pose health risks highlights the necessity for comprehensive water management and regular monitoring to assess water quality and mitigate potential health hazards [51]. This distribution underscores the complexity of maintaining ecological balance and ensuring water safety in an environment influenced by diverse human and natural factors.

## Environmental and health implications of pollution indicators and ecological impacts in Wadi Hanifah's waters

The environmental and health implications of the bacterial diversity in Wadi Hanifah are profound, calling for an in-depth analysis. This review synthesizes current literature to understand

the complex interplay between pollution indicators and their ecological and health impacts. The presence of *E. coli* across multiple sites in Wadi Hanifah is a classic hallmark of fecal contamination, potentially due to sewage leakage or agricultural runoff [52]. This finding is alarming due to the associated risks of waterborne diseases. Concurrently, *Enterobacter* and *Klebsiella* species, prevalent in the Wadi, indicate organic pollution likely stemming from urban runoff or suboptimal waste management [53]. These organisms highlight a significant human impact on water quality. Detecting *S. aureus* and *P. aeruginosa*, bacteria commonly linked to various human infections, underscores a public health emergency [54]. These pathogens, possibly entering the water through medical waste, pose a heightened risk in areas of human contact. Literature suggests that such bacterial dominance can severely disrupt the natural microbial ecology [55]. This disruption manifests in altered nutrient cycles and imbalances in the aquatic food chain, potentially leading to eutrophication, as noted by Grégoire et al. [56]. Detecting plant-associated bacteria like Erwinia points towards agricultural runoff, a source of fertilizers and pesticides. According to Mechan, this runoff compromises water quality and endangers aquatic life and crop safety. An alarming aspect brought forward in recent studies [57] is the potential rise in antibiotic-resistant bacteria due to these environmental conditions. This phenomenon poses a significant threat to global health and ecological sustainability. This literature review emphasizes the need for robust water treatment and management strategies, as echoed by researchers such as Ogidi et al. [58]. Effective interventions must focus on pollution control, sustainable waste management, and ecosystem preservation to safeguard public health and the ecological integrity of Wadi Hanifah.

This comprehensive study on Wadi Hanifah's water quality, faces limitations such as a constrained temporal scope that might not fully capture seasonal variations, and a limited spatial resolution, suggesting a need for more sampling sites to enhance granularity. Future research could benefit from incorporating a broader range of contaminants, such as emerging pollutants and microplastics, and employing advanced analytical techniques for a more in-depth understanding. Additionally, expanding the focus to include ecological impacts, socioeconomic aspects, and continuous monitoring systems would provide a more holistic view of the healthy water body and the effectiveness of pollution control strategies. These areas of future work are essential to build upon the current study's findings and address the complexities of water quality management in dynamic riverine systems.

## Conclusions

Comprehensive assessment of Wadi Hanifah's water quality reveals a multifaceted scenario impacted by physical, chemical, and biological factors. The study's findings, parallel to and diverging from existing literature, highlight significant spatial variations and potential pollution sources impacting the ecosystem. Elevated ammonia and phosphate levels, high sulfate, cyanide, and copper concentrations, and significant total hardness require urgent attention and management. The correlation of these findings with widespread bacterial presence underscores the interconnectedness of water quality parameters. This study emphasizes the necessity of continuous, multi-dimensional monitoring and targeted interventions to preserve the ecological balance and water quality of Wadi Hanifah. We recommend that the relevant authorities, such as the Ministry of Environment, Water and Agriculture, the Riyadh Development Authority, and the National Water Company, collaborate to implement effective pollution control measures, such as enforcing stricter regulations, enhancing wastewater treatment, promoting public awareness, and restoring natural habitats. We also suggest that the stakeholders, such as the local communities, industries, farmers, and researchers, engage in participatory water management and adopt sustainable water use practices to protect and conserve this valuable water resource.

## Supporting information

**S1 Fig. Natural water source's location in Wadi Hanifah.** L1-L10: Water sample before the filtration and water treatment; L11: Water sample after entering station door with simple filter; L12: Water sample after first water treatment; L13: Water sample after second water treatment, using filter, Fish and algae; and L14: Water sample after final filtration.
(DOCX)

## Author Contributions

**Conceptualization:** Hazem Aqel, Naif Sannan, Afnan Al-Hunaiti, Ramy Fodah.

**Data curation:** Hazem Aqel.

**Formal analysis:** Hazem Aqel.

**Investigation:** Hazem Aqel, Naif Sannan.

**Methodology:** Hazem Aqel, Naif Sannan.

**Project administration:** Hazem Aqel.

**Resources:** Afnan Al-Hunaiti, Ramy Fodah.

**Software:** Naif Sannan.

**Supervision:** Hazem Aqel.

**Validation:** Hazem Aqel, Naif Sannan, Afnan Al-Hunaiti, Ramy Fodah.

**Visualization:** Ramy Fodah.

**Writing – original draft:** Hazem Aqel.

**Writing – review & editing:** Naif Sannan, Afnan Al-Hunaiti, Ramy Fodah.

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
