## [Decision Letter · Decision Letter 0]

9 Jan 2024

PONE-D-23-42265Integrated Water Quality Dynamics in Wadi Hanifah: Physical, Chemical, and Biological PerspectivesPLOS ONE

Dear Dr. Aqel,

Thank you for submitting your manuscript to PLOS ONE. After careful consideration, we feel that it has merit but does not fully meet PLOS ONE’s publication criteria as it currently stands. Therefore, we invite you to submit a revised version of the manuscript that addresses the points raised during the review process.

We look forward to receiving your revised manuscript.

Kind regards,

Wajdy Jum’ah Al-Awaida, Ph.D

Academic Editor

PLOS ONE

Journal Requirements:

4. We note that Figure 1 in your submission contain map/satellite images which may be copyrighted. All PLOS content is published under the Creative Commons Attribution License (CC BY 4.0), which means that the manuscript, images, and Supporting Information files will be freely available online, and any third party is permitted to access, download, copy, distribute, and use these materials in any way, even commercially, with proper attribution. For these reasons, we cannot publish previously copyrighted maps or satellite images created using proprietary data, such as Google software (Google Maps, Street View, and Earth). For more information, see our copyright guidelines: http://journals.plos.org/plosone/s/licenses-and-copyright.

5. Please upload a copy of Figure 2, to which you refer in your text on page 11. If the figure is no longer to be included as part of the submission please remove all reference to it within the text.

Additional Editor Comments:

Dear Authers

After thoroughly reviewing the manuscript "Integrated Water Quality Dynamics in Wadi Hanifah: Physical, Chemical, and Biological Perspectives," I have identified several areas where the article could be significantly improved:

1. Inadequate Methodological Detail:

The description of the sampling methods and locations lacks sufficient detail, making it difficult to assess the replicability and rigor of the study.

The analytical techniques used for water quality assessment are not described in enough detail to evaluate their appropriateness or accuracy.

2. Data Presentation Issues:

The tables and figures, while present, are not effectively integrated into the text. Clearer references and explanations in the manuscript would better contextualize these visual aids.

The statistical analysis methods, if used, are not clearly outlined. There's a need for a more comprehensive explanation of the statistical methods and their rationale.

3. Weak Discussion and Conclusions:

The discussion lacks depth in comparing the findings with other studies or historical data. This limits the contextual understanding of the study's results.

The conclusions drawn are relatively generic and do not sufficiently encapsulate the specific findings of this study.

4. Lack of Broader Implications:

The manuscript does not adequately discuss the broader implications of its findings, particularly in relation to environmental management and policy.

5. Inconsistencies and Errors:

There are several inconsistencies in terminology and units of measurement, which can be confusing.

The manuscript contains grammatical and typographical errors that detract from its overall quality.

6. Limited Reference to Current Literature:

The references seem outdated or not sufficiently relevant to the study's focus, which raises questions about the thoroughness of the literature review.

7. Need for Additional Research Suggestions:

The study lacks suggestions for future research or fails to identify gaps in the current understanding that it addresses.

8. Ethical and Compliance Issues:

The manuscript does not clearly state ethical compliance related to sample collection, which is crucial for studies involving environmental sampling.

9. Funding and Conflict of Interest Statements:

The manuscript should include a clear statement regarding funding sources and any potential conflicts of interest.

In summary, while the manuscript provides valuable insights into the water quality of Wadi Hanifah, improvements in methodological clarity, data presentation, discussion depth, error correction, and literature contextualization are essential for enhancing its academic rigor and impact.

Reviewers' comments:

Reviewer's Responses to Questions

**Comments to the Author**

1. Is the manuscript technically sound, and do the data support the conclusions?

Reviewer #1: Yes

Reviewer #2: Yes

2. Has the statistical analysis been performed appropriately and rigorously? 

Reviewer #1: Yes

Reviewer #2: Yes

3. Have the authors made all data underlying the findings in their manuscript fully available?

Reviewer #1: Yes

Reviewer #2: Yes

4. Is the manuscript presented in an intelligible fashion and written in standard English?

Reviewer #1: Yes

Reviewer #2: Yes

5. Review Comments to the Author

Reviewer #1: The manuscript describes comprehensive evaluation of water quality in one of the known valleys in Arabian Peninsula of Saudi Arabia. Analysis included physical, Chemical constituents and bacterial contamination but did not discuss how the study's outcomes are different from the previous studies about the same valley. This point must be addressed carefully.

Reviewer #2: The author is required to follow the remarks provided in the report submitted to the journal editor, as these comments are essential for reaching a conclusive decision on whether to accept the publication of this article.

6. PLOS authors have the option to publish the peer review history of their article (what does this mean?). If published, this will include your full peer review and any attached files.

Reviewer #1: No

Reviewer #2: **Yes: **Husni othman

---

## [Author Response · Author response to Decision Letter 0]

16 Jan 2024

Subject: Response to Editorial and reviewer comments on Manuscript [PONE-D-23-42265]

Dear Dr. Al-Awaida,

Thank you for your letter regarding our manuscript entitled "[Manuscript Title]" submitted to PLOS ONE. We appreciate the opportunity to revise our manuscript and address the concerns raised by both you and the reviewers. We have carefully considered each point and have made appropriate revisions to the manuscript.

Revision in response to reviewer comments

We have addressed each of the specific comments made by the reviewers. Our responses are detailed in the attached 'Response to Reviewers' document. Here are some highlights of the revisions:

1. Introduction: Expanded details about the history and significance of Wadi Hanifa and how our study differs from previous investigations.

2. Materials and Methods: Provided more information on the selection of study locations, sample collection, and detailed descriptions of laboratory methods and principles.

3. Combine Results and Discussion

• Clarified the methodology for various tests and improved the presentation of data in tables and figures.

• Enhanced the depth of the discussion by comparing findings with other studies and addressed the implications of our findings.

Editorial comments and Journal requirements

1. Methodological details: We have provided additional details in the Methods section for better understanding and replicability of the study.

2. Data presentation: Revised tables and figures for better integration into the text and provided a clearer explanation of statistical methods used.

3. Discussion and Conclusions: Expanded the discussion section to provide more context and depth and refined the conclusions to be more specific to our findings.

4. Broader implications: Discussed the broader implications of our findings in relation to environmental management and policy.

5. Inconsistency and Errors: We have corrected all identified inconsistencies and errors.

6. Reference to current literature: Updated and expanded the list of references to include more relevant and recent studies.

7. Research suggestions and Ethical compliance: Included suggestions for future research and provided detailed information regarding ethical compliance.

8. Funding and conflict of interest statements: Updated the manuscript to include a clear statement regarding funding sources and any potential conflicts of interest.

9. Style and language editing: We have engaged a professional editing service to ensure that the manuscript meets PLOS ONE's style requirements and is free from language errors. The details of the service used, and the marked-up copy of the manuscript are included in the submission.

10. Figure copyright issues: We have addressed the concerns regarding Figure 1 by removing the figure.

11. Permits for field work: Included detailed information about the permits obtained for the study in the Methods section. (Lines 141-142)

We believe that these revisions address the concerns raised and significantly improve the manuscript. We hope that it now meets the publication criteria for PLOS ONE. We look forward to the opportunity to contribute to the journal.

1. Regarding Figure 1: We have already removed this figure from our manuscript, acknowledging the concerns regarding copyrighted maps or satellite images created using proprietary data.

2. Regarding S1 Figures: We would like to clarify that the S1 Figures are entirely original and do not require special permissions for publication under the CC BY 4.0 license. These images were captured by our research team directly from the collection sites. They are not sourced from any third-party proprietary software or databases, ensuring full compliance with the journal's copyright guidelines.

We hope this addresses your concerns satisfactorily. Please feel free to reach out if further clarification or action is needed.

Thank you for considering our work for publication in PLOS ONE. Please find the revised manuscript, along with the 'Response to Reviewers' document and other required items, attached.

Regards,

Hazem Aqel

---

## [Editor Report · Decision Letter 1]

22 Jan 2024

Integrated Water Quality Dynamics in Wadi Hanifah: Physical, Chemical, and Biological Perspectives

PONE-D-23-42265R1

Dear Dr. Aqel,

We’re pleased to inform you that your manuscript has been judged scientifically suitable for publication and will be formally accepted for publication once it meets all outstanding technical requirements.

Kind regards,

Wajdy Jum’ah Al-Awaida, Ph.D

Academic Editor

PLOS ONE
---

## [Editor Report · Acceptance letter]

7 Feb 2024

PONE-D-23-42265R1 

PLOS ONE

Dear Dr. Aqel, 

I'm pleased to inform you that your manuscript has been deemed suitable for publication in PLOS ONE. Congratulations! Your manuscript is now being handed over to our production team.

Kind regards, 

on behalf of

Prof. Wajdy Jum’ah Al-Awaida 

Academic Editor

PLOS ONE